# Process evaluation of a data-driven quality improvement program within a cluster randomised controlled trial to improve coronary heart disease management in Australian primary care

Nashid Hafiz[1]*, Karice Hyun[1,2], Qiang Tu[1], Andrew Knight[3,4], Charlotte Hespe[5], Clara K. Chow[6,7], Tom Briffa[8], Robyn Gallagher[9], Christopher M. Reid[10,11], David L. Hare[12], Nicholas Zwar[13], Mark Woodward[14,15], Stephen Jan[14], Emily R. Atkins[7,14], Tracey-Lea Laba[16], Elizabeth Halcomb[17], Tracey Johnson[18], Deborah Manandi[1], Tim Usherwood[7,14], Julie Redfern[1,14]

1 School of Health Sciences, Faculty of Medicine and Health, The University of Sydney, Camperdown, Australia, 2 Department of Cardiology, Concord Hospital, ANZAC Research Institute, Sydney, Australia, 3 Primary and Integrated Care Unit, Southwestern Sydney Local Health District, Sydney, Australia, 4 School of Public Health and Community Medicine, University of New South Wales, Sydney, Australia, 5 The University of Notre Dame, School of Medicine, Sydney, Australia, 6 Western Sydney Local Health District, Sydney, Australia, 7 Westmead Applied Research Centre, Faculty of Medicine and Health, University of Sydney, Sydney, Westmead, Australia, 8 School of Population and Global Health, The University of Western Australia, Perth, Australia, 9 Sydney Nursing School, Faculty of Medicine and Health, University of Sydney, Sydney, Australia, 10 School of Population Health, Curtin University, Perth, Australia, 11 School of Public Health and Preventive Medicine, Monash University, Melbourne, Australia, 12 University of Melbourne and Austin Health, Melbourne, Australia, 13 Faculty of Health Sciences & Medicine, Bond University, Gold Coast, Australia, 14 The George Institute for Global Health, University of New South Wales, Sydney, Australia, 15 The George Institute for Global Health, School of Public Health, Imperial College London, United Kingdom, 16 Clinical and Health Sciences, University of South Australia, Adelaide, Australia, 17 School of Nursing, University of Wollongong, Wollongong, Australia, 18 Inala Primary Care, Brisbane, Queensland, Australia

* nashid.hafiz@sydney.edu.au

**Data Availability Statement:** The study data cannot be publicly shared due to the ethical restrictions on sharing potentially identifiable

## Abstract

### Background

This study evaluates primary care practices' engagement with various features of a quality improvement (QI) intervention for patients with coronary heart disease (CHD) in four Australian states.

### Methods

Twenty-seven practices participated in the QI intervention from November 2019 –November 2020. A combination of surveys, semi-structured interviews and other materials within the QUality improvement in primary care to prevent hospitalisations and improve Effectiveness and efficiency of care for people Living with heart disease (QUEL) study were used in the process evaluation. Data were summarised using descriptive statistical and thematic analyses for 26 practices.

information, which may compromise the privacy of the participants, according to the NSW Population and Health Services Research Ethics Committee (CINSW). The participant consent form does not include public deposition of the data. Data is stored securely in the University of Sydney's Research Data Storage Database. However, in accordance with PLoS One guidelines, de-identified data can be made available upon request from the ethics committee (NSW Population & Health Services Research Ethics Committee). Contact information to request data: cinsw-ethics@health.nsw.gov.au.

**Funding:** Funding for this study was provided by a National Health and Medical Research Council (NHMRC) Partnership Project Grant (Award Grant Number: GNT1140807). Additional in-kind and cash support from the following partner organisations; Amgen (cash support), Austin Health, Australian Cardiovascular Health and Rehabilitation Association, Australian Commission on Safety and Quality in Health Care, Australian Primary Health Care Nurses Association, Brisbane South PHN, Fairfield General Practice Unit, Heart Support Australia, Improvement Foundation, Inala Primary Care, National Heart Foundation of Australia, Nepean Blue Mountains PHN (cash support), Royal Australian College of General Practitioners, Sanofi (provided cash support via the Externally Sponsored Collaboration pathway), South Western Sydney PHN, The George Institute for Global Health (cash support) and University of Melbourne. The funders, including funding body and industry partners had no role in study design, data collection and analysis, decision to publish, or preparation of the manuscript.

**Competing interests:** The funding body and industry partners were not involved in the design of the study and did not have any role during its execution, analyses, interpretation of the data, or decision to submit results. Amgen and Sanofi Australia have provided cash support to the main study. MW is a consultant to Amgen, Freeline and Kyowa Kirin. Other authors have nothing to disclose. This does not alter our adherence to PLOS ONE policy on sharing data and materials.

**Abbreviations: AP**, Activity period; **CDMPs**, Chronic Disease Management Plans; **CHD**, Coronary Heart Disease; **CVD**, Cardiovascular Disease; **GP**, General Practitioner; **IQI**, Interquartile interval; **LW**, Learning workshops; **PDSA**, Plan-Do-Study-Act; **PHN**, Primary Health Networks; **PIP-QI**, Practice Incentive Program-Quality Improvement; **PM**Practice Manager; **QI**, Quality Improvement; **QUEL**, QUality improvement in primary care to prevent hospitalisations and improve Effectiveness

## Results

Sixty-four practice team members and Primary Health Networks staff provided feedback, and nine of the 63 participants participated in the interviews. Seventy-eight percent (40/54) were either general practitioners or practice managers. Although 69% of the practices self-reported improvement in their management of heart disease, engagement with the intervention varied. Forty-two percent (11/26) of the practices attended five or more learning workshops, 69% (18/26) used Plan-Do-Study-Act cycles, and the median (Interquartile intervals) visits per practice to the online SharePoint site were 170 (146–252) visits. Qualitative data identified learning workshops and monthly feedback reports as the key features of the intervention.

## Conclusion

Practice engagement in a multi-featured data-driven QI intervention was common, with learning workshops and monthly feedback reports identified as the most useful features. A better understanding of these features will help influence future implementation of similar interventions.

## Trial registration

Australian New Zealand Clinical Trials Registry (ANZCTR) number ACTRN12619001790134.

## Introduction

Cardiovascular disease (CVD), including coronary heart disease (CHD), contributes to one-third of deaths annually and remains a significant contributor to disability globally [1, 2]. In Australia alone, CHD is responsible for 10% of all deaths and 41% of these are CVD-related deaths [3]. Primary care plays a crucial role in reducing the burden of CHD as it is the first point of contact for patients where their care is coordinated [4, 5]. Largely financed by the federal government's Medicare–a universal healthcare system, Australian primary care services that are available to patients with CHD include lifestyle counselling, prescription of guideline-recommended medications, chronic disease management plans (CDMPs) and participation in cardiac rehabilitation [6, 7]. In Australia, patients who had at least one follow-up with a General Practitioner (GP) or cardiologist or utilised any of the aforementioned services after an acute CHD event have been shown to lower the chance of emergency re-admission and death [8].

The Australian government introduced the Practice Incentive Program Quality Improvement (PIP-QI) in 2019 to enhance the management and quality of care provided to people with chronic disease [9]. To receive the incentives, primary care practices are required to participate in continuous quality improvement (QI) in partnership with their local Primary Health Networks (PHN) and submit quarterly data reports to the latter. PHNs are funded by the Australian government to work closely with individual practices to coordinate health services within local communities [10]. PHNs also provide feedback to practices and support capacity to perform QI activities to ensure optimal service delivery. However, more research is needed to understand how practices can best implement PIP-QI to improve care in CHD.

and efficiency of care for people Living with coronary heart disease; **SD**, Standard Deviation.

The availability of PIP-QI and advancing data collection and reporting systems have enabled practices to adopt data-driven QI programs [11, 12]. When implemented effectively, data-driven QI has demonstrated success in several health conditions, including diabetes, asthma, and COPD [13–15]. However, studies have identified that implementation and sustainability of such initiatives are complex and challenging [16]. To improve the management of chronic conditions, it is increasingly important to understand the features and processes associated with implementing such programs [17–19]. The "QUality improvement in primary care to prevent hospitalisations and improve Effectiveness and efficiency of care for people Living with heart disease (QUEL)" study is currently being conducted in Australia. The QUEL study protocol is published elsewhere [20]. QUEL included a multifaceted 12-month intervention aimed at improving the management of CHD care in primary care practices by using data-driven QI strategies. The primary objective of this study is to comprehensively evaluate the QUEL intervention by examining practice engagement in performing QI activities, providing insight into the delivery of the intervention, and assessing the usefulness of the intervention features from healthcare providers' perspective. Specifically, it aims to (i) describe and analyse practice engagement, time commitment, skills and capacity of the practice team members associated with the intervention and (ii) explore to what extent the intervention was delivered as intended and whether the intervention features were useful. We hypothesise that higher practice engagement and perceived usefulness of intervention features are positively associated with the increased adoption of data-driven QI strategies in improving the care of patients with CHD in primary care practices.

## Methods

### Study design

The QUEL study is a cluster randomised trial, where primary care practices were randomised to receive the QI intervention or continued to receive usual care without access to the intervention during the study. In addition to usual care, control practices were offered an opportunity to participate in a series of virtual workshops after the completion of 24 months data collection. For this study, a process evaluation was performed on the intervention practices using a mixed-methods approach, collecting both quantitative and qualitative data from 27 urban and rural primary care practices of varying sizes within ten PHNs and across four Australian states (New South Wales, South Australia, Victoria and Queensland). The protocol for the process evaluation is published elsewhere (S1 Appendix) [21]. Ethics approval was obtained from the New South Wales Population and Health Services Research Ethics Committee (HREC/18/CIPHS/44). Fig 1 provides a flow diagram of the process evaluation conducted.

### Participants

Participants were included if they met any of the following criteria: (i) team members from a practice randomised to receive the intervention, including general practitioners (GPs), nurses and practice managers (PM), (ii) PHN staff who provided direct support to intervention practices, and (iii) provided written informed consent.

### QI intervention

The QUEL intervention was delivered between November 2019 and November 2020. It consisted of multiple features, which included attendance at six learning workshops, monthly submission of data and plan-do-study-act (PDSA) cycles, receipt of monthly feedback reports and support from the study team or relevant PHNs. Learning workshops were delivered

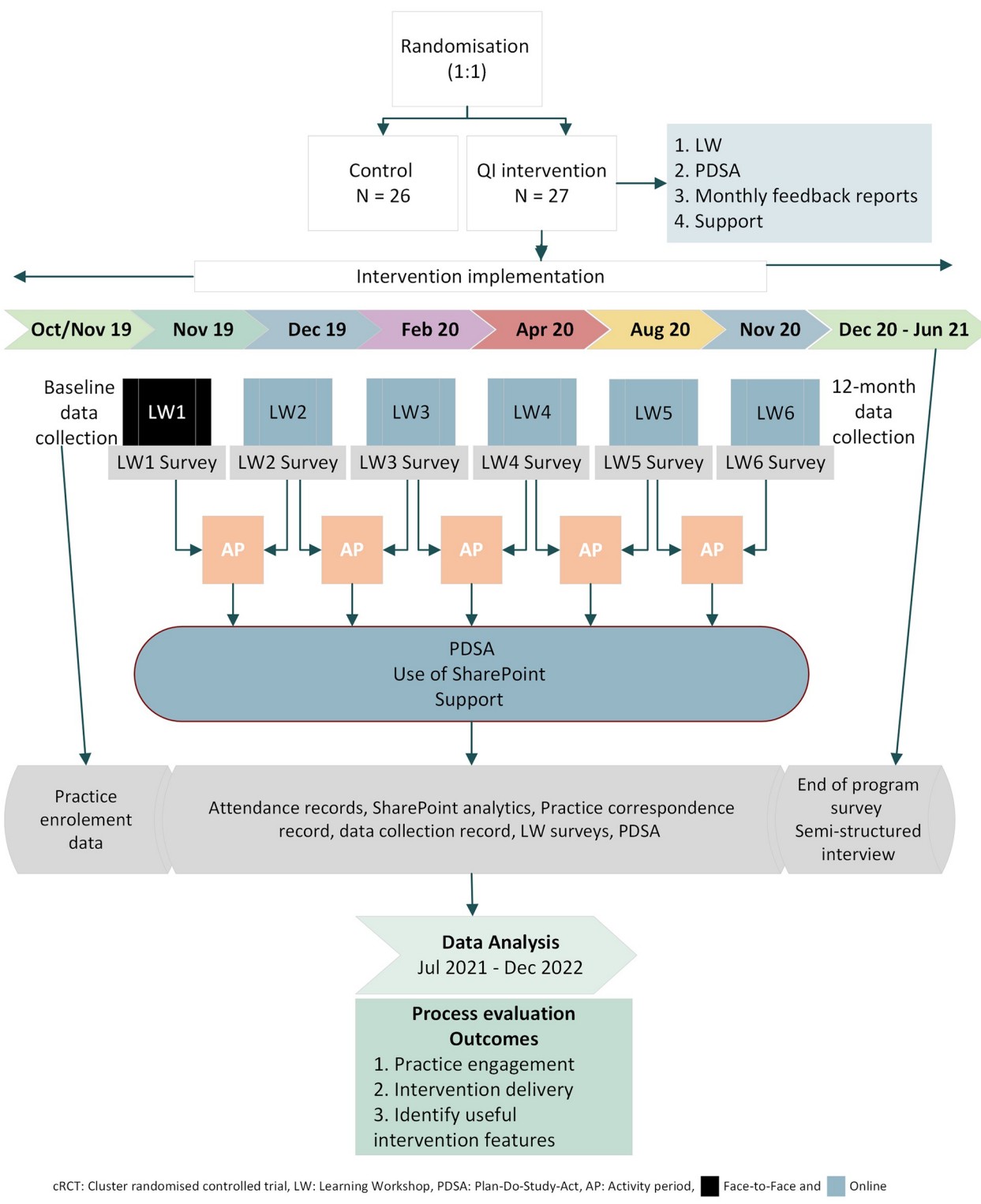

**Fig 1. Process evaluation flow diagram of QUEL intervention.**

approximately every two months, and the practice team members undertook other QI activities (i.e., electronic data submission, continuous improvement efforts, and feedback reports) between the learning workshops.

All practices and PHN staff were given an individual SharePoint account to access study materials, including workshop recordings, presentations and the QUEL handbook as intervention guidelines. Individual monthly reports and an online template to submit PDSAs were also provided in the account. After obtaining consent, PEN Computer Systems (PenCS) used their software to extract clinical data on the pre-defined CHD measures (S1 Table) from the intervention practices automatically each month and transmitted them to the study team [22].

After receiving electronic data, the study team reviewed and aggregated the data to create Excel reports and graphs, uploaded to the practice's individual SharePoint account as a PDF. These individual feedback graphs helped the practices to easily identify improvement areas and track their progress over time. Practices also used the PDSA cycles to test and implement changes. Each practice was required to submit monthly PDSA cycles focused on improving the pre-defined 12 performance measures for QUEL (S1 Table).

A vital component throughout the intervention was the external support provided to each practice by the study team or PHN staff. The study team and PHNs provided similar support to the practices, from ensuring practice participation in training and learning workshops, encouraging practice engagement, and helping with PDSA cycles to solving any data collection or feedback issues.

## Data sources

Data from the following sources were synthesised to evaluate the QI intervention and address the study aims: 1) practice-level enrolment data, 2) attendance record, 3) SharePoint resources, 4) practice correspondence record, 5) data collection record, 6) PDSA cycles, 7) learning workshop surveys, 8) end-of-program evaluation survey and 9) semi-structured interviews of practice team members and PHN Staff. Details of the data sources are published elsewhere [21]. These data were collected throughout the intervention period between November 2019 – November 2020. Additionally, the end-of-program evaluation survey and the semi-structured interviews were conducted at the intervention completion between December 2020 and June 2021. Feedback was sought from all practice team members who were involved in leading the QI activities. To ensure a balanced representation and minimise bias, team members for the semi-structured interviews were invited to participate from both rural and urban practices and from practices representing high, medium or low attendance in the learning workshops.

## Data protection and confidentiality

In accordance with ethical guidelines, the authors had access to information that could identify individual participants during the data collection phase. However, all identifiable information was removed and replaced with unique identification numbers. The data was treated with strict confidentiality and stored securely in the University's Research Storage Database. Access to the database was limited to the study team only, requiring a username and password.

## Outcome measures

Practice engagement with the QUEL intervention was defined as attendance in the series of learning workshops, submission of PDSA cycles and use of SharePoint by the practices. Workshop attendance data were collected after each of the six learning workshops (delivered online and face-to-face). PDSA cycle submission was collected in SharePoint, extracted and stored on a spreadsheet. SharePoint usage data was collected as part of the end-of-program evaluation

survey and using webpage analytics. Time commitment and skills of the practice team members were also collected using surveys.

The time commitment was measured as "Never(1)", "Rarely(2)", "Sometimes(3)", "Usually (4)", "Always(5)", and "Don't know (0)" using questions from the end-of-program evaluation survey. A score of ≥4/5 indicates longer time spent on QI activities. The semi-structured interviews also asked open-ended questions about the time spent by the team members on implementing these activities. The skills and capacity of a practice team member were defined as the roles, experience, and availability of practice team members leading the QI activities collected via surveys.

## Data analysis

One practice withdrew following participation in the first learning workshop due to staff change and was excluded from the analysis. Data from 26 practices were analysed for the process evaluation. Descriptive statistics were used to analyse quantitative data. Responses and measurements from all data sources are presented as numbers and percentages for categorical variables and mean and standard deviation (SD) or median and interquartile intervals (IQI) for continuous variables. Practices that did not respond to the survey were not included in the analysis. To evaluate the practice engagement, the workshop attendance, number of PDSA submissions, and SharePoint use were categorised into distinct groups for analysis. Workshop attendance was classified as low (less than three workshops attended), moderate (three to four workshops attended), and high (five or more workshops attended). PDSA submission was categorised as practices submitting less than three, three to six, and seven or more PDSA cycles. SharePoint use was grouped based on the number of visits made by the practices over the 12-month intervention period, with categories 0 to 149, 150 to 299, and 300 or more visits.

Qualitative data from semi-structured interviews, surveys and other data sources were analysed using thematic analysis [23]. Semi-structured interviews were conducted via Zoom, using an audio recorder and transcribed verbatim by NH and DM. Two researchers (NH and DM) performed thematic analysis of interview transcripts to ensure consistency in the interpretation of the themes. Both researchers individually prepared the data for transcribing, coded and reviewed them before defining the themes for interpretation [23, 24]. Minor disagreements about the interpretation of some responses and the categorisation of some themes were discussed with a third researcher (KH) until a consensus was reached. Free text from surveys and PDSAs were also coded thematically. Thematic analysis was performed using QSR NVivo version 1.6.1.

## Results

### Practice and PHN participation

Twenty-six primary care practices from four Australian states (69% of the practices from New South Wales, 15% from Victoria, 12% from South Australia and 4% from Queensland) participated in the QUEL intervention. Among these practices, six were from rural areas, and the remaining 22 were in urban areas across these four states. The practices were also of varying size, with the number of GPs varying from 1–18, and the median (IQI) number of GPs in these practices was 7 (6.25). Fifty-four team members from 26 primary care practices responded to at least one of the six learning workshop surveys, the end-of-program evaluation survey or participated in a semi-structured interview. Participants responding to each learning workshop survey ranged between 13 to 26. Thirty-six participants from 20 (77%) practices responded to the end-of-program evaluation survey. Eight team members from seven practices participated in the semi-structured interviews.

**Table 1. Summary of participants providing feedback on learning workshop surveys.**

| Learning workshops | No of practices | No of participants from practices | No of PHN | No of participants from PHN | Total participants provided feedback |
|---|---|---|---|---|---|
| *LW1 | 18 | 26 | 2 | 3 | 29 |
| LW2 | 16 | 20 | 3 | 3 | 23 |
| LW3 | 14 | 16 | 2 | 1 | 17 |
| LW4 | 11 | 14 | 1 | 1 | 15 |
| LW5 | 10 | 12 | 3 | 4 | 16 |
| LW6 | 17 | 26 | 3 | 3 | 29 |

*LW: Learning workshop, PHN: Primary Health Network

Five of the ten PHNs agreed to participate in the QUEL study; lack of time and capacity to undertake the additional responsibilities attributed to the non-participation of the remaining PHNs. Ten participants from these PHNs responded to the learning workshop surveys or participated in the semi-structured interviews. PHN staff were also encouraged to attend the learning workshops to track their practices' progress in the QI intervention. Four PHNs attended three (50%), and only one attended five (80%) learning workshops. The number of PHN staff attending the learning workshops ranged from one to four who also responded to the surveys. Only one PHN staff participated in the semi-structured interview. Table 1 summarises the number of participants providing feedback for the process evaluation at the end of each learning workshop.

**Practice engagement and attendance.** Fig 2 displays the distribution of practices across different levels of workshop attendance, PDSA cycles and SharePoint use over the 12-month intervention period in frequency graphs, providing detailed insights into the use of these features.

**Workshop attendance.** Forty-two percent (11/26) of the intervention practices attended five to six learning workshops, another 42% (11/26) attended three to four learning workshops, and only 16% (4/26) attended two or less learning workshops.

**PDSA submission.** Sixty-nine percent (18/26) of the intervention practices submitted 97 PDSA cycles over the intervention period. The average number of PDSA submissions per month ranged from one to 17. Twenty-one (22%) PDSAs were submitted without any associated dates. The median (IQI) PDSA submitted by the intervention practices over the 12 months was 3.5 (0, 6).

**Use of SharePoint.** Seventy percent (14/20) of the practices reported using the online account via the end-of-program evaluation survey during the one-year intervention period. From the SharePoint user analytics, we found the median (IQI) number of account visits per practice was 170 (106, 252) over the site's lifetime.

Table 2 reveals cross-tabulation of workshop attendance with PDSA submissions and SharePoint use, revealing a pattern in practice engagement. PDSA submission and SharePoint use were balanced for practices that attended five or more workshops. In contrast, practices that attended less than two workshops revealed low use.

Additionally, Table 3 provides a detailed summary of practice engagement with the intervention over the 12-months among participating practices. Overall, 11 practices that attended five or more learning workshops submitted 57 PDSAs, and the median (IQI) SharePoint use was 231 (150.5, 338.5). Another 11 practices that attended three to four workshops submitted 27 PDSAs, and the median (IQI) SharePoint use was 139 (110, 169.5). Four practices that attended less than three workshops submitted 13 PDSAs with a median (IQI) SharePoint use of 168.5 (91.5, 220.2).

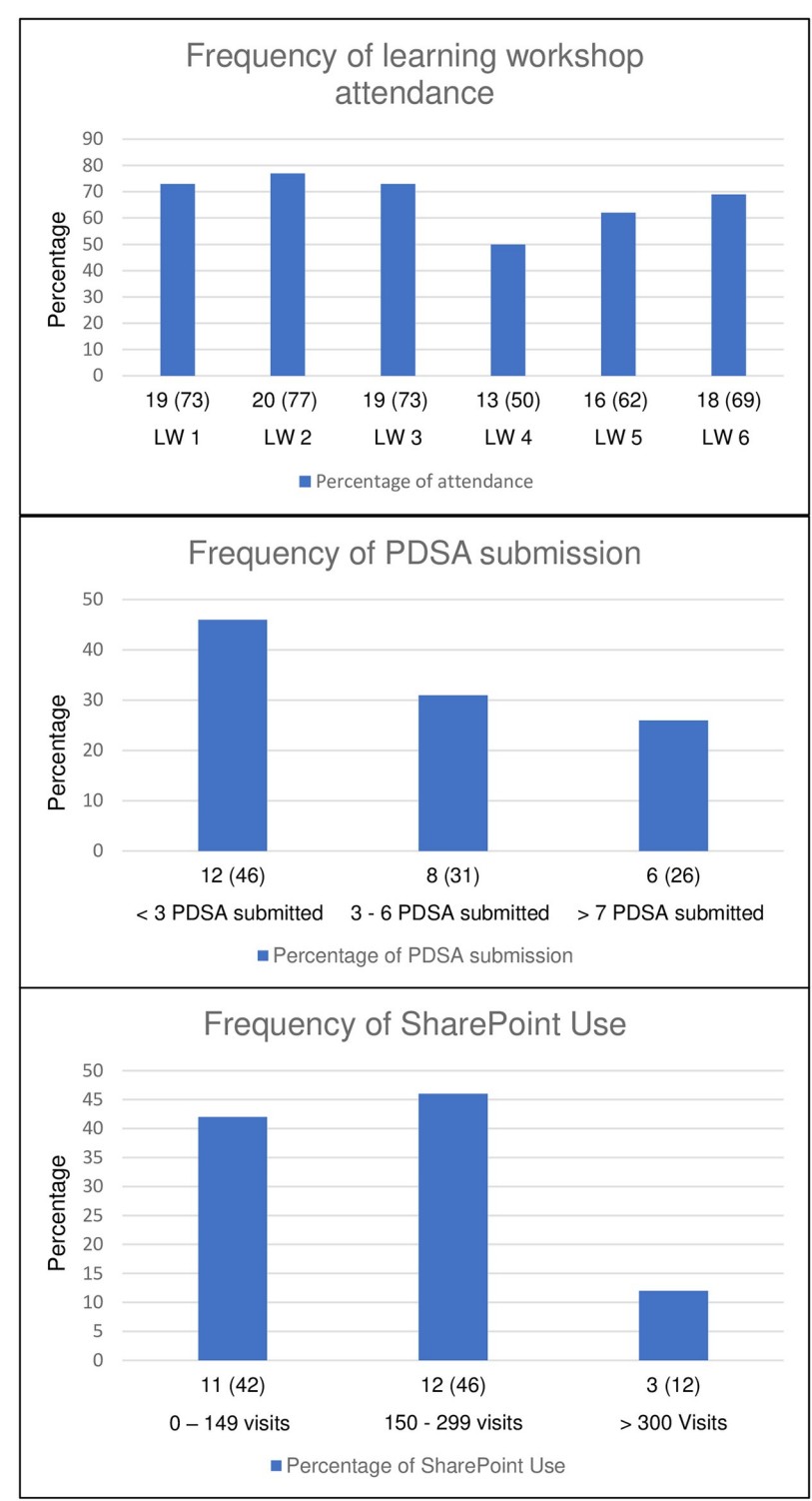

**Fig 2. Frequency distribution of practices' workshop attendance, PDSA submission and SharePoint use during the 12-month intervention period.**

Table 2. Workshop attendance, PDSA submission and SharePoint use practice distribution.

| Workshop attendance (n[a], %[b]) | PDSA submission (n) | | | SharePoint Use (n) | | |
|---|---|---|---|---|---|---|
| | < 3 | 3–6 | > 7 | 0–149 Visits | 150–299 Visits | > 300 Visits |
| 0–2 (4,16%) | 2 | 2 | 0 | 2 | 2 | 0 |
| 3–4 (11, 42%) | 7 | 3 | 2 | 7 | 5 | 0 |
| 5–6 (11, 42%) | 4 | 3 | 4 | 3 | 5 | 3 |

[a]n = is the number of practices in each category,

[b]Percentage is calculated = n/N, where N is the total no of practices in the intervention (26)

## Skills and capacity of the practice team members

As the intervention progressed, most practices designated team members to implement the QUEL QI changes within their practices. The majority of team members leading the QI activities were GPs (39%), followed by practice managers (PM) (39%) and nurses (18%). Others

Table 3. Detailed summary of practice engagement per practice.

| Practice Code | Number of workshops attended | PDSA submission | Sharepoint Use |
|---|---|---|---|
| A | 6 | 10 | 475 |
| B | 6 | 10 | 393 |
| C | 5 | 10 | 284 |
| D | 6 | 8 | 268 |
| E | 4 | 8 | 154 |
| F | 3 | 7 | 278 |
| G | 2 | 6 | 122 |
| H | 5 | 6 | 441 |
| I | 2 | 6 | 236 |
| J | 6 | 5 | 231 |
| K | 5 | 5 | 225 |
| L | 4 | 4 | 80 |
| M | 3 | 4 | 185 |
| N | 4 | 3 | 139 |
| O | 5 | 2 | 231 |
| P | 3 | 1 | 257 |
| Q | 5 | 1 | 50 |
| R | 1 | 1 | 215 |
| S | 3 | 0 | 154 |
| T | 3 | 0 | 101 |
| U | 4 | 0 | 119 |
| V | 6 | 0 | 76 |
| W | 4 | 0 | 131 |
| X | 6 | 0 | 65 |
| Y | 4 | 0 | 57 |
| Z | 0 | 0 | 0 |

Green: High engagement

Yellow: Moderate engagement

Red: Low engagement

**Table 4. Characteristics of practice team members leading the QI activities in the intervention practices.**

| Primary care Practices | |
|---|---|
| Number of participating primary care practices, n | 26 |
| Number of total participants providing feedback, n | 54 |
| Age, mean (SD) | 45.7 (11.8) |
| Female, n (%) | 37 (68) |
| **Health professional category, n (%)** | |
| GP/GP register/ Clinical Director/ Principal GP | 21 (39) |
| PM/ Assistant Practice Manager/ Practice Manager who is a nurse | 21 (39) |
| Practice Nurse/ Registered Practice Nurse/ Nurse Coordinator | 10 (18) |
| Other Admin and Research Officer | 2 (4) |
| **Years in the present position, n (%)** | |
| < 1 year | 4 (8) |
| 1–3 Years | 11 (21) |
| 3–5 Years | 6 (11) |
| > 5 Years | 31 (58) |
| Not reported | 1 (2) |

*GP: General Practitioner, PM: Practice manager

were research officers and admin staff (4%). More than half of the team members (58%) had five or more years of work experience. Thirty-one percent (17/54) of them were male, and the average age (SD) of the team members was 45.7 (11.8) years. Characteristics of practice team members are described in Table 4.

Team members who held leadership roles and had clinical backgrounds were able to take on the leadership and effectively drive changes in their practices, as described by one of the participants.

*"As the principal GP, I take on the leadership role. Whatever initiative we are undertaking as a practice, I explain to the staff, this is the reason why we are doing this, and then they will do it."*

(Practice K, Female, GP)

One participant described that they were unable to sustain QI activities within their practice due to not having a clinical background.

*"Disappointment from my point that I couldn't get it up and running because of not having the clinical background after the registrar left"*

(Practice W, Female, PM)

## Time commitment

Almost 70% (18/26) of the practices reported higher time commitment on using the electronic health system within working hours to identify CHD patients, monitor and track patients, develop care plans and record keeping. Table 5 presents a cross-tabulation of these practices on their engagement with the intervention. Additionally, two practices reported moderate

**Table 5. Practice engagement summary based on workshop attendance and time commitment.**

| Workshop attendance Category of attendance, (n[a], %[b]) | PDSA submission | SharePoint sse |
|---|---|---|
| 0–2 (1, 5%) | 6 | No of visits[c]—122 |
| 3–4 (9, 50%) | 24 | Median (IQI)—139 (119, 154) |
| 5–6 (8, 44%) | 42 | Median (IQI)—231 (188, 311)) |

[a]n = is the number of practices in each category,

[b]Percentage is calculated = n/N, where N is the total no of practices in the intervention who reported time commitment (20)

[c]No of visits were used as only one practice in the category.

level of time commitment to perform QI activities. Both practices attended five or more learning workshops, submitted 14 PDSAs, and the median (IQI) of SharePoint use was 268 and 441 respectively during the intervention period. 23% (6/26) of the practices that didn't report any data on time commitment showed varying levels of engagement. One practice attended five or more workshops, submitted one PDSA and visited the SharePoint site 50 times. Two practices attended three to four workshops, submitted 11 PDSAs, and the median (IQI) SharePoint use was 80 and 278 respectively. Three practices that attended less than three workshops submitted 7 PDSAs, and the median (IQI) of SharePoint use was 215 (108, 226).

Interview data revealed that most practices had weekly or monthly team meetings to track QI progress, and one practice reported having daily update meetings. Practice team members reported setting aside 5–10 minutes during those meetings to discuss the QI targets.

*"Our doctors have clinical meetings every Thursday and once a month, at the end of the meeting, I'd spend 5 minutes giving them an update and reminding them don't forget the QUEL project"*

(Practice B, Female, PM)

*"We have burst meetings where it would just be 5–10 minutes catching up on where we are at and what we need to do, and we stick to the plan. We only got three bullet points like what is working, what's not working, and how can we achieve the level of we want to achieve for the day."*

(Practice X, Female, PM)

Aside from regular meetings, practice team members set aside half an hour to half a day to perform QI activities.

*"We would normally put in 30 minutes to an hour a week to do recalls, reminders, data cleansing, etc."*

(Practice J, Female, PM)

*"Especially I worked on a Thursday. Thursday afternoons are always very quiet. So that gave me the best time to do stuff. I'd say half a day a week"*

(Practice B, Female, Nurse)

## Intervention delivered as intended, key intervention features and its usefulness

Learning workshop was identified as a key intervention feature by 60% of the practices, and one-third of the practices identified monthly feedback reporting as another important feature. Practice team members found these two features to be the most useful in facilitating QI changes within their practices. Qualitative data identified themes describing the usefulness of individual intervention features reported by the practice team members, illustrated in Box 1.

**Learning workshops.**   The first and the sixth learning workshops were initially planned as face-to-face events. Only the first learning workshop was delivered face-to-face before the impact of the COVID-19 pandemic [25], and the remaining five were all delivered virtually due to the ongoing restrictions. These workshops were scheduled approximately two months apart, but learning workshops five and six were approximately four months apart as the practices were busy with COVID-19 protocols and vaccinations.

**Electronic data submission and monthly feedback reports.**   Although all 26 practices submitted data and received monthly feedback reports most of the time as intended, there were exceptions in some months (S2 Table). The most common reason for not submitting data and receiving monthly reports was technical errors. Common technical errors were: (i) an error in the automatic data extraction system, (ii) the automatic data extraction system was turned off, and (iii) data extraction team not having access to technical support from the practice to run automated data collection. Once the technical issue was identified, the QUEL study team worked with relevant PHNs and the PenCS team to resolve the issue.

**PDSA cycles.**   The practice team members also received training in the learning workshops on implementing PDSA cycles. Further support was also available for the practice team members during the activity periods by the respective PHNs and study team. We anticipated practices submitting one PDSA cycle per month, in total, 12 cycles per practice over one year. Despite the training and support, only 12% (3/26) of the practices submitted ten or more and 57% (15/26) of the practices submitted between one to nine PDSAs during the one-year intervention (Table 2), suggesting the intervention was not implemented as intended.

**Support.**   Five PHNs that were participating in the QUEL study provided support to 12 of the 26 practices. Two of the five PHNs were located in New South Wales, which supported seven practices in that state. The other three PHNs were in South Australia, Queensland, and Victoria. The PHN in South Australia supported another three practices in that state, while the PHNs in Queensland and Victoria each supported only one practice within their respective regions. The remaining 14 practices were supported by the QUEL study team. Based on the support provided by the study team, the mean (SD) number of contacts between practices and the study team was 14 (4.3). PHNs contacted the practices independently, however data on PHN contacts were not collected during the intervention period. Contacts were made via phone calls, emails and in person to provide support to the practices to solve any technical errors, provide monthly updates, encourage practices to perform QI and help with any other queries. This support was useful in maintaining practices' engagement with the intervention features.

As a result of their participation in the QI intervention, practices reported the role of the practice team members changed during the one-year intervention. Around half (14/26) of the practices acknowledged an increase in the scope of their team members' roles to perform QI activities, such as data collection, coding, analysis, review, and reporting to meet QUEL targets. However, five practices reported no significant changes as QI was already a part of their role. Practice team members also performed various QI activities during the intervention year (S3 Table). Sixty-nine percent (18/26) of the practices reported an improvement in their quality of

Box 1. Quotes illustrating why practices found each intervention features useful.

| | Themes | Quotes |
|---|---|---|
| **Learning Workshops** | (i) Opportunities to learn from other practices | "*It was one practice, a country practice, and I can't remember the name; it was a tiny, small practice with the receptionist on board with the whole thing. I was just blown away by the way that they actually had embraced this project and done it.*" (Practice W, Female, PM)<br>"*There was a lot of collaboration such as sharing of experience in these workshops, which was quite helpful. I think just hearing the way that different practices had tackled certain tasks was quite helpful.*" (Practice V, Female, Nurse) |
| | (ii) Opportunities to share experiences with peers | "*We presented our Healthy Heart Clinic in one of the online learning workshops to other practices*" (Practice W, Female, PM) |
| | (iii) Regular get together to keep practices updated and reminded them to reinforce QI | "*By having a routine training or a catch-up or a meeting with a specific focus it brings us back to what we're aiming to do, particularly for the CHD.*" (Practice X, Female, PM)<br>"*Most important one is the workshops, I believe, important very, very important. Keep us updated all the time.*" (Practice Y, Female, GP) |
| **Monthly feedback report** | (i) To identify gaps and areas of improvement | "*We actually looked at all of our reports. We worked out from the graphs which were the lowest ones of the parameters. So, using the 12 measures and using that graph was very useful. Because it actually gave us our shortfall.*" (Practice W, Female, PM)<br>"*It's not my opinion or someone else's opinion; its actually data, and you can say, look we have only got 60% of our patients that have had a blood pressure in the last 12 months, who are on antihypertensive, we need to do better than tha*t." (Practice V, Female, Nurse) |
| | (ii) To track progress with QI | "*Well, I guess the monthly reports you provided kept us on our toes in a way, I guess. We could see how we were going easily within the project, so I think that was good.*" (Practice B, Female, Nurse)<br>"*Some of the doctors were quite shocked, in terms of some of the original results received from the monthly reports, and so it's a helpful thing to be able to have everyone going towards certain goals.*" (Practice V, Female, Nurse) |
| **PDSA** | (i) Helped to improve the quality of data | "*Our data has improved in small proportion in most areas of the 12 CHD measures for QUEL*" (Practice D, Female, Nurse) |
| | (ii) Helped to create awareness for correct coding of data | "*Clinic Drs have reported increased understanding of the need to "code" uniformly within the practice*" (Practice M, Female, GP) |
| | (iii) Produced successful outcome following a recall | "*We had one patient respond to an SMS for a blood test and also came in for a care plan.*" (Practice J, Female, PM) |

| Support | Build effective relationships between the study team and the practice | "*Your team visiting us physically, I feel pretty good, that means we are an important practice to visit, it improved the relationship between us*" (Practice Y, Female, GP) |
| | Provide training and support on using data extraction tools | "*If I'm having problems with the data extraction tool, he (practice support officer) will help me fix it*" (Practice B, Female, Nurse) |

*PM: Practice manager, CHD: Coronary heart disease, GP: General practitioner, SMS: Short message service, QI: Quality Improvement

care for CHD patients due to these activities. Only one practice reported no change, and another practice was uncertain of any changes in their quality of care. Seventy-three percent (19/26) of practices reported participating in QUEL enhanced their capacity to be PIP-QI ready. At the end of the intervention, 42% (11/26) of practices reported they were able to claim QI-PIP.

## Discussion

This study elaborates on the primary care practices' engagement with the QI intervention in improving care of CHD. The Intervention led to increased scope of QI activities for the practice team members leading to more than half of the practices (69%) self-reporting improvement in their QI activities for CHD patient care. As a result, majority (73%) of the practices felt they were ready for PIP-QI and some (42%) were able to claim the benefits. However, the practice engagement with the intervention was varied. Engagement with the PDSA cycles was low with only 12% of practices submitting ten or more cycles over 12 months. However, the submission range of PDSA cycles was diverse (0–10) in all practices despite the varied attendance. Practices that attended a higher number of learning workshops submitted higher number of PDSA cycles and showed higher engagement with the SharePoint site. However, variations were seen within each attendance category, indicating several factors contributing to the different engagement level. Additionally, practices reporting higher time commitment generally demonstrated higher learning workshop attendance, submitted more PDSAs, and used SharePoint more, suggesting a positive correlation between time commitment and engagement in the intervention with some variations in a small portion of the practices. Qualitative analysis identified team members in a leading role or with clinical backgrounds were able to implement QI changes more effectively within their practices and practices regularly set aside additional time during working hours to implement these changes. The study also identified learning workshops and monthly feedback reports as the two key useful intervention features to facilitate QI activities and changes.

Several QI strategies are currently being practiced in clinical settings including the Model for improvement, Lean and Six Sigma [26]. The Model for Improvement, used in this intervention is a widely used QI strategy in healthcare, which provides a systematic approach for planning, testing and implementing changes [27]. In addition, learning workshops, PDSA cycles, feedback reports and support were also used in combination as the QI intervention to improve CHD care [28–31]. Findings from our study suggest that higher attendance in learning workshops could have positively influenced PDSA submission and SharePoint use, indicating its significance in increasing practice engagement to perform QI activities. However, associations between the intervention features and improvement in clinical outcomes have yet

to be established. It is important to evaluate the effect of QI intervention on improving outcomes as research found, practices receiving regular feedback reports were able to improve clinical outcomes, particularly in achieving better blood pressure control in patients with hypertension [32] and risk factor screening [33, 34]. While a systematic review revealed mixed findings on the effectiveness of PDSAs on improving clinical outcomes [35], other studies identified several factors influencing the reduced engagement level [36, 37], similar to our findings. Our study also included a variety of skilled professionals including GPs, nurses or PMs as practice team members to lead the QI changes within their practices, which is also an important strategy for successful implementation of QI interventions [37–39]. Lastly, while some studies demonstrated the importance of using practice support within QI strategies to improve care [40–42], the current study did not provide a deeper understanding of the level of support required for successful implementation of such programs.

A strength of this study is the use of mixed-methods research, a commonly used method in evaluating QI studies, as it has the ability to strengthen data quality and provide a robust interpretation of the results [36, 42, 43]. Further, we also combined both qualitative and quantitative data and performed triangulation of the multiple datasets providing a wide range of perspectives from multiple health professionals, consequently providing an in-depth understanding of the complex intervention features. The practices included in the evaluation were from various sizes and regions ensuring wider representation of participants, therefore enhancing the generalisability of the findings to similar healthcare settings. The intervention features and implementation strategies described in this study can be used as a useful framework to be replicated with modifications in similar healthcare settings aiming to improve the quality of care for patients with chronic diseases.

The study, primarily focused on exploring the efficacy of the intervention rather than the effectiveness, has several limitations that may influence its potential wider roll-out into practice. The intervention period coincided with the outbreak of the global COVID-19 pandemic, potentially impacting practices, attendance in learning workshops and overall engagement [25]. Other limitations were the potential introduction of response and reporting bias arising from some practices not responding to the surveys and reliance on self-reported data, respectively. The exclusion of non-responsive practices from the quantitative analysis and continuous checking of data for errors and accuracy were used to reduce the bias. Additionally, categorical data was collected to measure health professionals' time spent on delivering the QI program, but continuous data would have offered a more accurate measure of time spent. Further, almost half of the practices were supported by their respective PHNs. However, due to the independent operations of the PHNs, we were unable to obtain comprehensive data regarding the extent of support provided by both the PHNs and the study team during the intervention period. Finally, it was beyond the scope of the process evaluation to evaluate whether the intervention was effective in improving the pre-defined CHD. These limitations may affect the generalizability of our findings and the feasibility of implementing the intervention on a larger scale.

Findings from the study suggest that external factors, unexpected events or occurrences should be taken into consideration when planning broader implementation strategies. We also acknowledge that the paper could benefit from a more detailed exploration of PDSAs in understanding a direct association between workshop attendance and PDSA submissions with the intervention. Highlighting a scope for future research to explore the various factors associated with the PDSA engagement [44]. Adding a more robust, sophisticated, and accurate data collection and analysis method could provide more nuanced measures of the findings, therefore enhancing the feasibility of future studies. The limitation in accurately assessing the level of support required for the successful implementation can be addressed by establishing an

effective collaboration and incorporating remote reporting and data collection between the research team and the PHNs [36]. Furthermore, enabling tailored support and addressing the nuanced dynamics of time commitment is important to optimise engagement with QI interventions across diverse practices. The use of these combined strategies, along with ongoing training, designating clinicians as QI champions and increased use of data-driven technology to monitor progress, can collectively contribute towards the large-scale roll-out of future QI programs with an aim to improve care for patients with CHD across diverse healthcare settings [45, 46].

## Conclusion

The study highlights the varied engagement of primary care practices with the QI intervention aimed at improving the care of CHD. Learning workshops, monthly feedback reports, and PDSA cycles were found to be useful features of the intervention. Successful implementation of the intervention also depended on the additional time commitment and efforts of the practice team members, particularly GPs, nurses and practice managers, towards implementing QI changes within their practices. These findings offer valuable insights that can support other primary care practices seeking future adoption of these evolving data-driven QI initiatives, ultimately leading to improved patients' outcomes and more effective management of CHD and other chronic diseases. However, as healthcare continues to evolve in utilising data, further research is needed to evaluate the intrinsic factors influencing practices' engagement in such complex interventions and obtain a comprehensive understanding of how these strategies can be best implemented.

## Supporting information

**S1 Appendix. Published protocol.**
(PDF)

**S1 Table. The 12 CHD measures for QUEL study.**
(DOCX)

**S2 Table. Monthly data extraction submitted and feedback report received.**
(DOCX)

**S3 Table. Quotes summarising quality improvement activities performed by practices.**
(DOCX)

**S1 Checklist. PLOS ONE clinical studies checklist.**
(DOCX)

**S2 Checklist. STROBE statement—Checklist of items that should be included in reports of observational studies.**
(DOCX)

## Acknowledgments

The authors acknowledge the support of all the PHN and primary care practices who continue to support the QUEL project. Also, PenCS for providing the services and eHealth data platform for the study; and the Improvement Foundation for their continuous support in the delivery of the QI program and other study partners including; Inala Primary Care, Fairfield Hospital General Practice Unit, Australian Primary Health Care Nurses Association, Royal Australian College of General Practitioners, Australian Commission on Safety and Quality in

Health Care, Heart Support Australia Ltd, Austin Health, Australian Cardiovascular Health and Rehabilitation Association, National Heart Foundation, Sanofi, and Amgen. The authors would also like to acknowledge the ongoing contribution of Kane Williams in the legal arrangement and Caroline Wu in the research management of the trial.

## Author Contributions

**Conceptualization:** Nashid Hafiz, Julie Redfern.

**Formal analysis:** Nashid Hafiz, Karice Hyun.

**Methodology:** Nashid Hafiz, Karice Hyun, Qiang Tu, Julie Redfern.

**Resources:** Nashid Hafiz.

**Supervision:** Karice Hyun, Julie Redfern.

**Visualization:** Nashid Hafiz.

**Writing – original draft:** Nashid Hafiz.

**Writing – review & editing:** Qiang Tu, Andrew Knight, Charlotte Hespe, Clara K. Chow, Tom Briffa, Robyn Gallagher, Christopher M. Reid, David L. Hare, Nicholas Zwar, Mark Woodward, Stephen Jan, Emily R. Atkins, Tracey-Lea Laba, Elizabeth Halcomb, Tracey Johnson, Deborah Manandi, Tim Usherwood, Julie Redfern.

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
