## [Decision Letter · Decision Letter 0]

3 Nov 2023

PONE-D-23-21449Process evaluation of a data-driven quality improvement program within a cluster randomised controlled trial to improve coronary heart disease management in Australian primary carePLOS ONE

Dear Dr. Hafiz,

Thank you for submitting your manuscript to PLOS ONE. After careful consideration, we feel that it has merit but does not fully meet PLOS ONE’s publication criteria as it currently stands. Therefore, we invite you to submit a revised version of the manuscript that addresses the points raised during the review process.

We look forward to receiving your revised manuscript.

Kind regards,

Anandakumar Haldorai, PhD

Academic Editor

PLOS ONE

Journal Requirements:

"Funding for this study was provided by a National Health and Medical Research Council (NHMRC) Partnership Project Grant (Award Grant Number: GNT1140807). Additional in-kind and cash support from the following partner organisations; Amgen (cash support), Austin Health, Australian Cardiovascular Health and Rehabilitation Association, Australian Commission on Safety and Quality in Health Care, Australian Primary Health Care Nurses Association, Brisbane South PHN, Fairfield General Practice Unit, Heart Support Australia, Improvement Foundation, Inala Primary Care, National Heart Foundation of Australia, Nepean Blue Mountains PHN (cash support), Royal Australian College of General Practitioners, Sanofi (provided cash support via the Externally Sponsored Collaboration pathway), South Western Sydney PHN, The George Institute for Global Health (cash support) and University of Melbourne. JR is funded by a NHMRC Investigator Grant (GNT1143538). KH is supported by the NHMRC Investigator Grant (Emerging leadership 1) (APP1196724). MW is supported by the NHMRC grants (1080206 and 1149987). CR is supported by a NHMRC Principal Research Fellowship (APP1136372). TL is funded by a NHMRC Early Career Fellowship (APP110230). EA is supported by a National Heart Foundation Australia postdoctoral fellowship (101884). CC’s salary is funded by a Career Development Fellowship level 2 co-funded by the NHMRC and National Heart Foundation Future Leader Award (APP1105447), which supports 0.05FTE for trial meetings."

"The funding body and industry partners was not involved in the design of the study; and will not have any role during its execution, analyses, interpretation of the data, or decision to submit results. Amgen and Sanofi Australia has provided cash support to the main study. MW is a consultant to Amgen, Freeline and Kyowa Kirin. Other authors has nothing to disclose."

Additional Editor Comments:

Please carefully address the issues raised in the comments and, up front in your revised paper. Your revised paper will be sent to the same reviewers, as well as possibly new reviewers, for evaluation.

Make sure the Abstract briefly describes the paper as it is used in abstracting and citation services. Clearly specify the Purpose, Methodology, and problem findings.

Highlight the major spatial modulation objective in section 1. What is your work contribution in this area.

Spell out each acronym the first time used in the body of the paper. Spell out acronyms in the Abstract only if used there.

You may ignore any suggestion of including self-references by reviewers if not applicable.

Make sure that the Conclusion briefly summarizes the results of the paper it should not repeat phrases from the Introduction. Keep the Conclusion to about 200 words. Do not use any references or acronyms in the Conclusion.

Make sure all figures and tables are referred to in the body of the paper.

Tabulate the major drawbacks and problem identification under section 2. Also relate the problem with your proposed approach.

Explain the section 3 with proper diagrammatic approach.

In section 4, Give some real time industrial case study approach to elaborate your proposed approach.

It is recommended to use a professional proofread and native English correction. Papers with less than excellent English will not be published even if technically perfect.

Remove all unwanted and self-citations from the reference section.

Reviewers' comments:

Reviewer's Responses to Questions

**Comments to the Author**

1. Is the manuscript technically sound, and do the data support the conclusions?

Reviewer #1: Yes

Reviewer #2: Partly

2. Has the statistical analysis been performed appropriately and rigorously? 

Reviewer #1: Yes

Reviewer #2: No

3. Have the authors made all data underlying the findings in their manuscript fully available?

Reviewer #1: Yes

Reviewer #2: No

4. Is the manuscript presented in an intelligible fashion and written in standard English?

Reviewer #1: Yes

Reviewer #2: Yes

5. Review Comments to the Author

Reviewer #1: The article is well written with detailed statistics of the process evaluation but I found that the major limitation was a lack of actionable new information that would be useful to clinicians or researchers. It may be helpful to emphasise this more in the discussion.

Reviewer #2: This paper aims to report the process evaluation of a data-driven quality improvement program as part of a trial (the QUEL program) to improve the management of coronary heart disease (CHD) in primary care in Australia. The paper has the potential to make a useful contribution because we need to know more about the best ways to implement interventions in the complicated world of primary healthcare. However, the paper is not clear enough about its main focus, does not report the statistics in sufficient detail, and lacks information about key aspects of the process that would have implications for the wider roll-out of the intervention beyond the trial (information which you might expect to get from a process evaluation). These things make it hard to be sure that the conclusions drawn are warranted by the findings.

Focus

It is important to separate the presumed primary outcome of the trial, which I understand to be a positive change in the management of CHD as evidenced by the 12 measures given in the S1 Table, from the focus of the process evaluation reported in this paper. An improvement in CHD management is predicted to arise from iterative improvements in these 12 measures using a data-driven quality improvement paradigm. In other words, the purpose of the trial was to encourage primary care practices to use quality improvement methods to improve their performance on these measures successively over a 12-month period. Thus a key aspect of the process evaluation would be to understand whether the practices did indeed focus on these data and make improvements. However, the outcome measures for this paper were defined as attendance at workshops, submission of PDSA cycles and use of SharePoint (p.9). Time commitment and skills of the practice team members were also recorded, but there do not appear to be any outcomes that are clearly related to the use of data to drive the quality improvement. It is possible that the submission of PDSA cycles is the proxy for this, but the paper does not describe any examples of how practices designed PDSA cycles to make improvements in the measures, and the data for workshop attendance is not linked to the data on PDSA submission (i.e., we do not know whether practices with higher attendance at workshops submit more PDSA cycles).

So, the paper needs to be clearer about which processes are being evaluated and the measures being used.

Statistics

I found much of the presentation of descriptive statistics to be rather unsophisticated and, as suggested above, much more could be learnt by some cross-tabulation of the frequencies. So, submission of PDSA cycles and use of analytics could have been broken down by attendance at workshops, and staff category could be cross-tabulated with years of experience and gender. In the text accompanying each table, there tends to be a lot of repetition of the data that is already reported in the table which is unnecessary and tedious to read. Cross-tabulation would allow more nuanced comments to be drawn about the relationships and differences. There is also a lack of detail and precision in reporting some data, e.g., the size of primary care practices is described as 'varying', data gathering was conducted "using a variety of sources including learning workshop surveys, attendance records, PDSAs, among others", interview participants "were selected based on factors such as gender, location, and practice performance". In each of these cases, more detail needs to be provided. In addition the use of range categories reduces the level of information. For instance, in Table 2 why not use a frequency chart to display how many practices attended 1,2,3, etc. workshops? Or similarly to display the number of practices submitting different numbers of PDSA cycles over the 12 month period?

In the qualitative data, each quote should be attributed to a practice ID, as well as the gender and staff category. This is so that we know that the quotes come from a range of practices.

The extra sophistication of analysis and detail would need to be reflected in the points made in the Discussion section.

Implications for wider roll-out

The trial that is the subject of this process evaluation appears to be an efficacy trial rather than an effectiveness trial and although there are several occasions when the authors describe the high level of support provided by the trial team, they also conclude "we were unable to obtain comprehensive data regarding the extent of support provided by both the PHNs and study team during the intervention period". Clearly, an important part of the process evaluation would be to understand whether a high level of support is going to be needed to introduce the intervention on a wider scale and ensure fidelity with the intervention; a level of support which may be unrealistic. However, this paper is not able to report on this, which rather weakens its contribution. At the very least, the authors might have considered the implications of this lack of information.

Minor points

line 127 - it is not clear what is meant by receiving usual care here.

lines 136-9 - are these inclusion criteria additive or alternatives?

lines 157-8 - what are these data?

lines 177-80 - more detail on these data items would be useful.

line 232 - what level of disagreement was there about themes?

6. PLOS authors have the option to publish the peer review history of their article (what does this mean?). If published, this will include your full peer review and any attached files.

Reviewer #1: No

Reviewer #2: No

---

## [Author Response · Author response to Decision Letter 0]

3 Dec 2023

Response: Formatting checked throughout the manuscript. 

Response: The financial information “Funding for this study was provided by a National Health and Medical Research Council (NHMRC) Partnership Project Grant (Award Grant Number: GNT1140807). Additional in-kind and cash support from the following partner organisations; Amgen (cash support), Austin Health, Australian Cardiovascular Health and Rehabilitation Association, Australian Commission on Safety and Quality in Health Care, Australian Primary Health Care Nurses Association, Brisbane South PHN, Fairfield General Practice Unit, Heart Support Australia, Improvement Foundation, Inala Primary Care, National Heart Foundation of Australia, Nepean Blue Mountains PHN (cash support), Royal Australian College of General Practitioners, Sanofi (provided cash support via the Externally Sponsored Collaboration pathway), South Western Sydney PHN, The George Institute for Global Health (cash support) and University of Melbourne” has been included in the cover letter to be updated.

3. Thank you for stating the following financial disclosure: “Funding for this study was provided by a National Health and Medical Research Council (NHMRC) Partnership Project Grant (Award Grant Number: GNT1140807). Additional in-kind and cash support from the following partner organisations; Amgen (cash support), Austin Health, Australian Cardiovascular Health and Rehabilitation Association, Australian Commission on Safety and Quality in Health Care, Australian Primary Health Care Nurses Association, Brisbane South PHN, Fairfield General Practice Unit, Heart Support Australia, Improvement Foundation, Inala Primary Care, National Heart Foundation of Australia, Nepean Blue Mountains PHN (cash support), Royal Australian College of General Practitioners, Sanofi (provided cash support via the Externally Sponsored Collaboration pathway), South Western Sydney PHN, The George Institute for Global Health (cash support) and University of Melbourne. JR is funded by a NHMRC Investigator Grant (GNT1143538). KH is supported by the NHMRC Investigator Grant (Emerging leadership 1) (APP1196724). MW is supported by the NHMRC grants (1080206 and 1149987). CR is supported by a NHMRC Principal Research Fellowship (APP1136372). TL is funded by a NHMRC Early Career Fellowship (APP110230). EA is supported by a National Heart Foundation Australia postdoctoral fellowship (101884). CC’s salary is funded by a Career Development Fellowship level 2 co-funded by the NHMRC and National Heart Foundation Future Leader Award (APP1105447), which supports 0.05FTE for trial meetings."

Response – The following statement, “Role of Funder - The funders, including funding body and industry partners, had no role in study design, data collection and analysis, decision to publish, or preparation of the manuscript.” has been included in the cover letter to be updated. 

"The funding body and industry partners was not involved in the design of the study; and will not have any role during its execution, analyses, interpretation of the data, or decision to submit results. Amgen and Sanofi Australia has provided cash support to the main study. MW is a consultant to Amgen, Freeline and Kyowa Kirin. Other authors havenothing to disclose."

Response – The following “Competing Interest Statement - The funding body and industry partners were not involved in the design of the study and did not have any role during its execution, analyses, interpretation of the data, or decision to submit results. Amgen and Sanofi Australia have provided cash support to the main study. MW is a consultant to Amgen, Freeline and Kyowa Kirin. Other authors have nothing to disclose. This does not alter our adherence to PLOS ONE policy on sharing data and materials” has been included in the cover letter to be updated. 

Response – The following “Data availability statement: The study data cannot be publicly shared due to the ethical restrictions on sharing potentially identifiable information, which may compromise the privacy of the participants, according to the NSW Population and Health Services Research Ethics Committee (PHSREC). The participant consent form does not include public deposition of the data. Data is stored securely in the University of Sydney’s Research Data Storage Database. However, in accordance with PLoS One guidelines, de-identified data can be made available upon request from the ethics committee (PHSREC). Contact information to request data: cinsw-ethics@health.nsw.gov.au.” has been included in the cover letter to be updated. 

Response – Ethics statement from page 35 has been deleted. 

Additional Editor Comments:

Please carefully address the issues raised in the comments and, up front in your revised paper. Your revised paper will be sent to the same reviewers, as well as possibly new reviewers, for evaluation.

7. Make sure the Abstract briefly describes the paper as it is used in abstracting and citation services. Clearly specify the Purpose, Methodology, and problem findings.

Response - Abstract has been updated throughout. Please see pages 3-4, lines 55 – 81 of the revised manuscript. 

8. Highlight the major spatial modulation objective in section 1. What is your work contribution in this area. 

Response – The following sentences have been edited and added in section 1 - “The primary objective of this study is to comprehensively evaluate the QUEL intervention by examining practice engagement, providing insight into the intervention and delivery and assessing the usefulness of the intervention features from healthcare providers perspective. Specifically, it aims to (i) describe and analyse practice engagement, time commitment, skills and capacity of the practice team members associated with the intervention; and (ii) explore to what extent the intervention was delivered as intended, whether the intervention features were useful. We hypothesise that higher practice engagement and perceived usefulness of intervention features are positively associated with the adoption of data-driven QI strategies in improving the care of patients with CHD in primary care practices.” Please see page 6, lines 124 – 136 of the revised Manuscript. 

9. Spell out each acronym the first time used in the body of the paper. Spell out acronyms in the Abstract only if used there.

Response – Acronyms have been spelled out the first time used in the body of the paper. Acronyms have been spelled out in the Abstract, only if used there. Please see pages 3-4, lines 55-81 of the revised manuscript. 

10. You may ignore any suggestion of including self-references by reviewers if not applicable.

Response – Not applicable; no reviewers requested self-references. 

11. Make sure that the Conclusion briefly summarizes the results of the paper it should not repeat phrases from the Introduction. Keep the Conclusion to about 200 words. Do not use any references or acronyms in the Conclusion.

Response – The conclusion has been updated to summarise the results of the paper. Please see pages 33 - 34, lines 630 - 647 of the revised manuscript.

12. Make sure all figures and tables are referred to in the body of the paper.

Response – All figures and tables have been referred to throughout the body of the manuscript. 

13. Tabulate the major drawbacks and problem identification under section 2. Also relate the problem with your proposed approach.

Response – Challenges faced during the study and their proposed solutions are discussed in details within section 4 as limitations. Please see pages 31 - 32, lines 586 – 607 of the revised manuscript.

14. Explain the section 3 with proper diagrammatic approach. 

Response – A flow diagram of the process evaluation has been developed using VISIO and has been referred as Fig 1 on page 7 of the revised manuscript. Please see attached Fig 1. 

15. In section 4, Give some real time industrial case study approach to elaborate your proposed approach.

Response – Case studies have been added throughout section 4. Please see pages 28 – 30, lines 516 – 567 of the revised manuscript.

16. It is recommended to use a professional proofread and native English correction. Papers with less than excellent English will not be published even if technically perfect.

Response – Professional proofreading (Grammarly) and native English correction have been used in the manuscript. 

17. Remove all unwanted and self-citations from the reference section.

Response – All unwanted and self-citations have been removed from the reference section. 

Reviewers' comments:

Reviewer's Responses to Questions

Comments to the Author

1. Is the manuscript technically sound, and do the data support the conclusions?

Reviewer #1: Yes

Reviewer #2: Partly

2. Has the statistical analysis been performed appropriately and rigorously?

Reviewer #1: Yes

Reviewer #2: No

Have the authors made all data underlying the findings in their manuscript fully available?

Reviewer #1: Yes

Reviewer #2: No

Is the manuscript presented in an intelligible fashion and written in standard English?

Reviewer #1: Yes

Reviewer #2: Yes

3. Review Comments to the Author

Reviewer #1: The article is well written with detailed statistics of the process evaluation but I found that the major limitation was a lack of actionable new information that would be useful to clinicians or researchers. It may be helpful to emphasise this more in the discussion.

Reviewer #2: This paper aims to report the process evaluation of a data-driven quality improvement program as part of a trial (the QUEL program) to improve the management of coronary heart disease (CHD) in primary care in Australia. The paper has the potential to make a useful contribution because we need to know more about the best ways to implement interventions in the complicated world of primary healthcare. However, the paper is not clear enough about its main focus, does not report the statistics in sufficient detail, and lacks information about key aspects of the process that would have implications for the wider roll-out of the intervention beyond the trial (information which you might expect to get from a process evaluation). These things make it hard to be sure that the conclusions drawn are warranted by the findings.

but I found that the major limitation was a lack of actionable new information that would be useful to clinicians or researchers. It may be helpful to emphasise this more in the discussion.

Response: Section 4 (Discussion) has been edited to include new information. Please see pages 27 – 33, lines 489 – 627 of the revised manuscript. 

Focus

It is important to separate the presumed primary outcome of the trial, which I understand to be a positive change in the management of CHD as evidenced by the 12 measures given in the S1 Table, from the focus of the process evaluation reported in this paper. An improvement in CHD management is predicted to arise from iterative improvements in these 12 measures using a data-driven quality improvement paradigm. In other words, the purpose 

---

## [Decision Letter · Decision Letter 1]

31 Jan 2024

Process evaluation of a data-driven quality improvement program within a cluster randomised controlled trial to improve coronary heart disease management in Australian primary care

PONE-D-23-21449R1

Dear Dr. Hafiz,

We’re pleased to inform you that your manuscript has been judged scientifically suitable for publication and will be formally accepted for publication once it meets all outstanding technical requirements.

Kind regards,

Anandakumar Haldorai, PhD

Academic Editor

PLOS ONE

Additional Editor Comments (optional):

Recommended

Reviewers' comments:

Reviewer's Responses to Questions

**Comments to the Author**

1. If the authors have adequately addressed your comments raised in a previous round of review and you feel that this manuscript is now acceptable for publication, you may indicate that here to bypass the “Comments to the Author” section, enter your conflict of interest statement in the “Confidential to Editor” section, and submit your "Accept" recommendation.

Reviewer #2: All comments have been addressed

2. Is the manuscript technically sound, and do the data support the conclusions?

Reviewer #2: Yes

3. Has the statistical analysis been performed appropriately and rigorously? 

Reviewer #2: Yes

4. Have the authors made all data underlying the findings in their manuscript fully available?

Reviewer #2: Yes

5. Is the manuscript presented in an intelligible fashion and written in standard English?

Reviewer #2: No

6. Review Comments to the Author

Reviewer #2: Regarding my answer to Q5 - The manuscript is fine in terms of intelligibility and use of English, in fact it is very good. However, I spotted a few minor errors. In the Results section of the Abstract, there is an error in the number of practice members being quoted; it says sixty-three, when it should be fifty-three. In addition there is an unfinished statement saying "Forty-two percent (11/26) of the practices attended five or more", but five or more what? This, I think, should say "...five or more learning workshops". Figure 1 probably needs a key, for the acronyms e.g., LW, AP, PDSA. The Glossary of acronyms after the Conclusion should be in alphabetical order.

Otherwise, I commend the authors for responding to the revisions in an excellent manner. In my humble opinion, this is a much better paper now.

7. PLOS authors have the option to publish the peer review history of their article (what does this mean?). If published, this will include your full peer review and any attached files.

Reviewer #2: **Yes: **Peter H. Gardner

---

## [Editor Report · Acceptance letter]

4 May 2024

PONE-D-23-21449R1 

PLOS ONE

Dear Dr. Hafiz, 

I'm pleased to inform you that your manuscript has been deemed suitable for publication in PLOS ONE. Congratulations! Your manuscript is now being handed over to our production team.

Kind regards, 

on behalf of

Dr. Anandakumar Haldorai 

Academic Editor

PLOS ONE